# Al_2_O_3_-Based a-IGZO Schottky Diodes for Temperature Sensing

**DOI:** 10.3390/s19020224

**Published:** 2019-01-09

**Authors:** Qianqian Guo, Fei Lu, Qiulin Tan, Tianhao Zhou, Jijun Xiong, Wendong Zhang

**Affiliations:** 1Key Laboratory of Instrumentation Science & Dynamic Measurement, Ministry of Education, North University of China, Tai Yuan 030051, China; qqguo0214@163.com (Q.G.); lufei_55@163.com (F.L.); 18734580862@163.com (T.Z.); xiongjijun@nuc.edu.cn (J.X.); wdzhang163com@163.com (W.Z.); 2Science and Technology on Electronic Test and Measurement Laboratory, North University of China, Tai Yuan 030051, China

**Keywords:** high-temperature, Schottky diodes, Al_2_O_3_, a-IGZO

## Abstract

High-temperature electronic devices and sensors that operate in harsh environments, especially high-temperature environments, have attracted widespread attention. An Al_2_O_3_ based a-IGZO (amorphous indium-gallium-zinc-oxide) Schottky diode sensor is proposed. The diodes are tested at 21–400 °C, and the design and fabrication process of the Schottky diodes and the testing methods are introduced. Herein, a series of factors influencing diode performance are studied to obtain the relationship between diode ideal factor *n*, the barrier height Ф*_B_*, and temperature. The sensitivity of the diode sensors is 0.81 mV/°C, 1.37 mV/°C, and 1.59 mV/°C when the forward current density of the diode is 1 × 10^−5^ A/cm^2^, 1 × 10^−4^ A/cm^2^, and 1 × 10^−3^ A/cm^2^, respectively.

## 1. Introduction

A harsh environment usually involves factors such as extreme temperature, high pressure, and high shock. High-temperature electronics and sensors that can operate between 21 °C and 400 °C have drawn considerable attention owing to their wide applications in harsh environments. Integrated sensing modules that can operate at high temperatures will be beneficial for many industrial applications. A high-temperature integrated circuit is an important part of such systems, so it must comprise diodes and transistors that can operate at high temperatures. The simplest device that can be integrated with a circuit is based on semiconductor diodes [1].

It has already been proved that the application of some wide-bandgap semiconductor materials overcome the defects of silicon materials, making semiconductor devices capable of operating at temperatures above 300 °C [2]. Shao et al. [3] prepared a 4H-SiC-based P-N junction diode using Ni/Ti and Al/Ti/Ni, respectively, with N- and P- 4H-SiC forming an ohmic contact with a forward conduction voltage of 2.6 V at room temperature and 1.4 V at 500 °C at an average voltage drift rate of 2.2 mV/K. Dipalo et al. describe a novel ISFET structure monolithically integrated with an InAlN/GaN HEMT structure [4]. Pearton et al. prepared a sensor based on GaN and can operate at 400 °C [5]. In addition to SiC, gallium nitride (GaN) is also widely used at harsh environments [6,7,8,9]. Gregory et al., [10] at the University of Rhode Island in the United States, proposed a thin-film temperature sensor based on a Schottky barrier diode that can be used to test the surface temperature of an ultra-high temperature blade at 25–1100 °C. Among the available oxide semiconductor films, amorphous indium-gallium-zinc-oxide (a-IGZO) is a preferred material for thin-film transistor (TFT) semiconductor layers because of its uniformity, compactness, the largest channel mobility, reproducibility, high electron mobility, simple fabrication process, good stability, and shows the most commercial potential [11,12,13,14,15].

Further, a-IGZO is a good amorphous semiconductor material as the internal physical properties are isotropic. For a-IGZO with *n* > 10^17^ cm^−3^, when the temperature changes, the carrier concentration is basically unchanged, indicating that the Fermi level (EF) has crossed the mobility edge. The Hall voltage signal can be detected in the a-IGZO film with *n* > 10^16^ cm^−3^, indicating that the carrier is not localized [16]. However, in the range of 10^16^ cm^−3^ < *n* < 10^17^ cm^−3^, the carrier concentration is independent of temperature, but Hall mobility still exhibits thermal activation characteristics, which means that there is a barrier above the mobility side [16,17]. Reducing the reverse-bias leakage current of the Schottky diode increases sharply to the extent of thermal runaway with increasing temperature. To date, most research on metal oxide semiconductors has focused on TFTs [18,19], while research on diodes is scarce [15,20,21]. Therefore, a-IGZO is ideal for the electrical materials used in harsh environments, especially at high temperature, due to its excellent electrical and physical properties and wide bandgap energy (3.5 eV).

In this study, an a-IGZO Schottky diode functioning at high temperature is demonstrated and experimentally characterized. The diode demonstrates stable operation within a temperature range of 21 °C to 400 °C. The fabricated a-IGZO Schottky diode has a turn-on voltage of 0.64 V to 0.14 V, with temperature rising from 21 °C to 400 °C.

## 2. Design and Fabrication

### 2.1. Diode Design

Here, we demonstrate what, to the best of our knowledge, is the first high-temperature a-IGZO Schottky diode fabricated at room temperature and operating at 21–400 °C. First, we fabricated Schottky diodes with different Schottky contact metals, a-IGZO layer thicknesses, and oxygen/argon ratios during the sputtering deposition process of the a-IGZO film. The structure of the proposed a-IGZO Schottky diode with a vertical design is illustrated in Figure 1 [22]. Next, we analyzed their current–voltage (J–V) characteristics at room temperature. Second, the I–V characteristics were tested at 21–400 °C in a vacuum environment using Lakeshore Model CRX-6.5K (Lake Shore Cryotronics, Inc. Ohio, USA) and the dependence of turn-on voltage and equivalent resistance of the Schottky diode on the temperature was obtained by analyzing the J–V characteristics.

### 2.2. Equivalent Model of the Diode and Extracted Parameters

For fabrication and testing at much higher temperatures, we used Al_2_O_3_ ceramic as the substrate to fabricate the a-IGZO Schottky diodes to study the effects of some key parameters, such as direct current (DC), on the device. An aluminum (Al) electrode is used here as the ohmic contact because the work function of Al is only 4.2 eV and the contact resistance between Al and a-IGZO is known to be quite low [23]. The Aurum (Au) and platinum (Pt) electrodes are chosen as the Schottky contact due to their high work functions of 5.1 and 5.65 eV, respectively [24]. We measured the contact resistance between Al and a-IGZO and the value is 1.191 Ωcm^2^. The overlapping region between the two electrodes determines the effective area of the diode. The Au/Pt Schottky contact area is modified by photolithography to obtain a device with an effective area of 200 μm × 200 μm and 300 μm × 300 μm. Figure 2a shows the cross-sectional schematic of the a-IGZO diodes [25], L_S_ represents the thickness of the a-IGZO layer and W_D_ is the depletion width caused by the Schottky barrier. The following four a-IGZO thicknesses (L) were investigated: 30, 50, 100, and 150 nm. An equivalent circuit of the diode is shown in Figure 2b, where R_S_ is the series resistance and consists of the bulk resistance of the a-IGZO layer and the ohmic contact resistance of the Al contact. To describe high-frequency characteristics, an effective series resistance R_ES_ was introduced, which was obtained by subtracting the equivalent resistance of the depletion region from R_S_ [26].

The a-IGZO semiconductor is an N-type semiconductor. The ideal energy band before contact with a certain metal is shown in Figure 3a, and the vacuum level is used as the reference energy level, where Ф*_m_* is the metal’s work function, Χ is the electron affinity, and Ф*_s_* is the semiconductor’s work function. Figure 3b shows the energy band diagram of the Schottky junction in which the N-type semiconductor is in contact with the metal. Usually, Ф*_m_* > Ф*_s_*, and at this time, the Fermi level of the N-type semiconductor is higher than that of the metal. In order to achieve equilibrium of the respective Fermi levels, electrons flow from the semiconductor to the metal, and the transfer of charges causes the interface area of semiconductor and metal to deplete free carriers and the interface area, also known as the depletion layer [26,27].

The ideal barrier height of the semiconductor after the metal is brought in contact with the N-type semiconductor is represented as Ф_*B*0_, and electrons move from the metal to the semiconductor to form a Schottky barrier, as represented by Equation (1).

(1)$${Ф}_{B0}=\left({Ф}_{m}-X\right)$$

As represented by Equation (2), *V_bi_* is the built-in voltage, and this barrier is formed for the electrons in the conduction band moving toward the metal, similar to the junction barrier.
(2)$$V_{bi}=\left({Ф}_{B0}-{Ф}_{n}\right)$$

According to the theory of thermionic emission, the voltage applied to both sides of the metal–semiconductor barrier region and the density of the current passing through the rectifying junction can be represented by Equation (3):
(3)$$J=A^{*}T^{2}\exp\left(\frac{-q{Ф}_{B}}{kT}\right)\left[\exp\left(\frac{qV_{a}}{nkT}\right)-1\right]$$where *k* is the Boltzmann constant (1.3806505 × 10^−23^ J/K), *T* is the thermodynamic temperature, and *A** is the effective Richardson constant, which can be obtained from the effective electron mass in the semiconductor material. For the a-IGZO material in this paper, m* = 0.34m_0_, where m_0_ is the mass of a single free electron and its value is 9.109 × 10^−31^ kg; thus, m* = 41 A/cm^2^ × K^2^.

### 2.3. Device Fabrication

This experiment used Al_2_O_3_ ceramic as a substrate for thin-film diode preparation. Before the fabrication, the substrates were degreased by supersonic cleaning in deionized water, acetone, and methanol, and then dried with nitrogen. Ti-Pt electrodes (10–50 nm thick) were deposited by RF sputtering at 80 W in pure argon, and subsequently, a-IGZO was formed by doping ZnO, Ga_2_O_3_, and In_2_O_3_ in a certain proportion (In/Ga/Zn = 1:1:1) and was sputtered at 80 W with 1% argon and 4% oxygen at 0.5 Pa. Al electrodes (50 nm thick) were deposited by RF sputtering at 80 W in pure argon. Patterns were defined by photolithography with the standard processes. All devices were fabricated at room temperature without any thermal annealing process. The J–V characteristics were obtained using a Keithley 4200A-SCS semiconductor analyzer and Lakeshore Model CRX-6.5K versatile cryogen-free micro-manipulated probe station at the temperature range of 20 K to 675 K.

In this study, oxygen vacancies were considered to be the source of carriers for a-IGZO films. However, too many oxygen vacancies can lead to many internal defect states in the active layer and affect device performance [25,28,29]. Therefore, we adjusted the oxygen-argon ratio in the sputtering environment during the sputtering process and the film thickness, and incorporated a certain amount of oxygen to eliminate excessive oxygen vacancies [30,31,32] to create better performance devices.

## 3. Results and Discussion

### 3.1. a-IGZO Schottky Diodes on Al_2_O_3_ Ceramics

The Schottky contact between the anode metal and the semiconductor layer directly affects the rectification performance of the diode. Figure 4 shows the optical images of the fabricated device with different positive electrode material.

The J–V characteristics of the a-IGZO Schottky diodes on Al_2_O_3_ ceramics with different Schottky contact metals are shown in Figure 5. From the Figure 5a, the rectification junction area is 300 μm × 300 μm, the diode with an Au and a Pt Schottky contact metal exhibited a rectification ratio of 100.4 and 119.4, respectively. At an applied voltage of 1 V, the diode with a 50-nm-thick a-IGZO layer between the Au and Al contacts exhibited a current density of 0.074 mA/cm^2^ and that between the Pt and Al contacts exhibited a current density of 1.20 mA/cm^2^. From Figure 5b, the rectification junction area was 200 μm × 200 μm, the Au/a-IGZO rectification ratio was 25.1, and the Pt/a-IGZO rectification ratio was 42.7. At an applied voltage of 1 V, the current density of Au and Pt Schottky contacts were 0.034 mA/cm^2^ and 19.9 mA/cm^2^, respectively. The diode with a Pt electrode exhibited a higher rectification ratio because when the metal and semiconductor were in contact, the energy band of the semiconductor at the interface was bent to form a Schottky barrier. The larger the metal work function, the larger the interfacial resistance and the higher the rectification ratio. Although the reverse-bias leakage current of the Au/a-IGZO diode is relatively low, its rectification ratio is much lower than that of the Pt/a-IGZO diode. Considering the above factors, we chose Pt as the Schottky metal.

The series resistance has a sensitive dependence on the thickness of the a-IGZO layer. The J–V characteristics of the a-IGZO Schottky diodes on Al_2_O_3_ ceramics with different a-IGZO thicknesses are shown in Figure 6. As shown in Figure 6a, for the diodes with 100 nm, 50 nm, and 30 nm a-IGZO layer thicknesses, the current densities are 0.042, 1.19, and 15.2 mA/cm^2^ and the rectification ratios are 887.2, 94.8, and 28.1, respectively. From Figure 5b, for the diodes with 150, 100, and 50 nm a-IGZO layer thicknesses, the current densities are 66.3, 0.005, and 20.1 A/cm^2^ and the rectification ratios are 79.1, 44.1, and 233.9, respectively. Taking Figure 6b as an example, given that the effective Richardson constant of a-IGZO is 41 A cm^2^ K^2^ and based on the thermionic emission theory [28,33,34], the extracted barrier heights are 0.41, 0.47, and 0.50 eV for diodes with 50, 100, and 150 nm a-IGZO layer thicknesses, respectively. The slope of the linear segment of the curve shows that the diode’s ideal factor n with an a-IGZO film thickness of 50 nm is also the largest.

Oxygen vacancies are the main source of carriers in a-IGZO films. However, excessive oxygen vacancies can lead to several defect states in the film and affect the device performance. An excessive oxygen content in the a-IGZO sputtering deposition process will degrade the stability of the film and reduce the electron mobility. For the Schottky barrier diodes studied here, when a lower forward bias voltage is applied, the performance of the diode depends on the metal–semiconductor Schottky contact barrier rather than the electron mobility of the a-IGZO layer. Therefore, appropriate oxygen addition during sputtering is used to passivate the Schottky contact surface, thereby reducing the Fermi level pinning effect [35]. When the forward bias voltage is high, the on-state current of the diode is limited by the series resistance, which is closely related to the electron mobility of the a-IGZO film. The incorporation of oxygen in argon during the sputtering process can result in a trade-off between the requirements for Schottky contact high barriers and relatively low series resistance.

The surface morphology of the samples is analyzed using the AFM technique. The AFM images of 50-nm-thick a-IGZO films with oxygen/argon ratios of 1% and 4% are shown in Figure 7. It can be seen that when the oxygen content is 4%, the surface roughness of the 50-nm-thick a-IGZO film is slightly larger than that of the film particle with the oxygen content of 1%. However, a film having an oxygen content of 1% has poor continuity and uniformity, and has island-like particle clusters, which is not conducive for the formation of an ideal contact between the metal and the a-IGZO film.

The targets used for the sputtering were InGaZnO4 targets with a standard atomic ratio of In: Ga: Zn = 1:1:1. According to Olziersky et al. [36], the atomic ratio of In/Ga varies with the preparation conditions. When the atomic ratio is 0.95, the carrier mobility is high, from the EDS analysis shown in Table 1, when the oxygen content is 4%, the atomic ratio of In/Ga close to 0.95. The Figure 8a shows the XRD pattern of the prepared a-IGZO film on Al_2_O_3_ substrate. The result of XRD analysis showed that there is an amorphous diffuse peak when the diffraction angle is less than 20°, and no strong crystal diffraction peak is present. Since the thickness of the film is only 50 nm, the diffraction peak of the Al_2_O_3_ substrate is strong when the diffraction angle is greater than 20°. Strong diffraction peaks of Al_2_O_3_ substrate when the diffraction angle is greater than 20°. The prepared a-IGZO film is amorphous. Shown in Figure 8b is a cross-sectional view of the diode, and Figure 8c is a surface view of the a-IGZO film. The surface of Al_2_O_3_ has pits with a diameter of about 2 μm, but the thickness of Ti/Pt/a-IGZO/Al deposited on the surface is consistent with the actual device.

The J–V characteristics of the Pt/a-IGZO Schottky diodes on Al_2_O_3_ ceramics with different oxygen/argon ratios sputtering conditions are shown in Figure 9, where for the diodes with 1% and 4% oxygen/argon ratio sputtering processes, the current densities are 0.34 mA/cm^2^ and 20.13 mA/cm^2^ and the rectification ratios are 29.2 and 233.9, respectively. During the sputtering process, the appropriate amount of oxygen in the argon gas can fill in the vacancy of the a-IGZO layer, reduce the surface defects, reduce the equivalent series resistance of the contact surface, and increase the Schottky contact barrier height.

### 3.2. High-Temperature Properties

In this study, the I–V characteristics of the Pt/a-IGZO diode with a film thickness of 50 nm and the sputtering process of O_2_: (Ar + O_2_) = 4% were tested in the range of 21–400 °C. The starting temperature was room temperature (21 °C), and the sample was tested at intervals of 50 °C. The I–V and J-V characteristics of the a-IGZO-based diode in the temperature range 21–400 °C are shown in Figure 10.

It shows when the device is at the forward bias, it exhibits forward rectification characteristics as expected and the current increases with temperature for a given voltage at forward. At a lower bias, a new feature dominated by a leakage component adds non-linearity to the I-V plot. Bartolomeo and Giubileo et al. have found that Equation (4) provides a perfect fit with the I-V plot at a low bias [37].
(4)$$I=I_{O}\left[e^{\frac{q\left(V-R_{s}I\right)}{nKT}}-1\right]$$
where $I_{O}$ can be described as Equation (5):(5)$$I_{O}=AA^{*}T^{2}e^{-{Ф}_{B}/KT}$$

The saturation current $I_{O}$ was obtained by extrapolating the linear intermediate voltage region of the linear part of the curve to a zero applied bias voltage for each temperature. The experimental values of the barrier height (${Ф}_{B}$) and the ideality factor (n) for the Schottky diode were determined from intercepts and slopes of the forward bias lnI versus V plot at each temperature, they can be obtained from Equations (4) and (5) and are presented in Figure 11a. Figure 11b shows resistance extracted from Cheung functions [37].

The forward conduction voltage of the Pt/a-IGZO diode decreases with the temperature increasing from 21 °C to 400 °C; in addition, the voltage is reduced from 0.64 V to 0.14 V, the average voltage drift rate is 1.32 mV/°C, the turn-on resistance is reduced from 3.81 kΩ to 1.54 kΩ, and the average resistance drift rate is 5.99 Ω/°C. When the bias voltage is negative, reverse-bias leakage current also increases with increasing temperature, the reverse-bias leakage current increased from 3.44 × 10^−8^ A to 0.03 μA to 0.47 μA. The curve of the turn-on voltage is shown in Figure 12a, which shows that when the temperature increases, the turn-on voltage decreases, and the change law is approximately linear. The turn on-resistance also gradually decreases and is fitted with an exponential function as the temperature changes; the result is shown in Figure 12b.

With an increase in forward voltage, the voltage drop across the series resistance strongly limits the exponential increase of the current. The equivalent resistance of the diode decreases at high temperatures and the effect of resistance on the entire device is reduced, Ф*_B_* increases gradually with temperature, with a barrier height of 0.73 eV at 21 °C and 1.25 eV at 400 °C, and the ideal factor *n* gradually decreases with temperature (1.54 at 21 °C and 1.09 at 400 °C) [38].

It is used in the temperature sensor to extract the forward voltage of the diode with a forward current density of 1 × 10^−5^ A/cm^2^, 1 × 10^−4^ A/cm^2^, and 1 × 10^−3^ A/cm^2^. Figure 13a shows the relationship between temperature and forward voltage at different current density and the voltage decreases approximately linearly with the increase of temperature. In order to verify the repeatability of the sensor, the temperature was measured at different concentrations three times and is shown in Figure 13b. It can be found that the sensor’s response is approximately the same at the same current density, respectively, in each heating and cooling cycle. When the forward current density of the diode is 1 × 10^−5^ A/cm^2^, the sensitivity of the sensor is 0.81 mV/°C; when the forward current density is 1 × 10^−4^ A/cm^2^, the sensitivity is 1.37 mV/°C; when the forward current density is 1 × 10^−3^ A/cm^2^, the sensitivity is 1.59 mV/°C.

The predicted temperature inaccuracy is merely −4.5–+5.5 °C (20–150 °C), −10.7–+12 °C (150–300 °C), and −5.5–+9 °C (300–450 °C) using two-point calibration within the range of 20–450 °C.

## 4. Conclusions

So far, we studied the reliability of the device, including the variation of diode with IGZO layer thickness and sputtering oxygen concentration. We have demonstrated that Pt/a–IGZO Schottky diodes on Al_2_O_3_ ceramic substrates with 50 nm a-IGZO thickness and a sputtering oxygen concentration of O_2_:(O_2_ + Ar) = 4% can operate in the temperature range of 21–400 °C. The forward conduction voltage of the Pt/a–IGZO diode decreased from 0.64 V at 21 °C to 0.14 V at 400 °C, and the average voltage drift rate was 1.32 mV/°C. The Ф_B_ increased gradually with increasing temperature from 0.91 eV at 21 °C to 1.67 eV at 400 °C, the ideal factor *n* gradually decreased from 1.06 at 21 °C and 0.59 at 400 °C. When it is used as a temperature sensor, the sensitivity of the sensor is 0.81 mV/°C, 1.37 mV/°C, and 1.59 mV/°C when the forward current density of the diode is 10^−5^ A/cm^2^, 10^−4^ A/cm^2^, and 10^−3^ A/cm^2^, respectively.

## Figures and Tables

**Figure 1 sensors-19-00224-f001:**
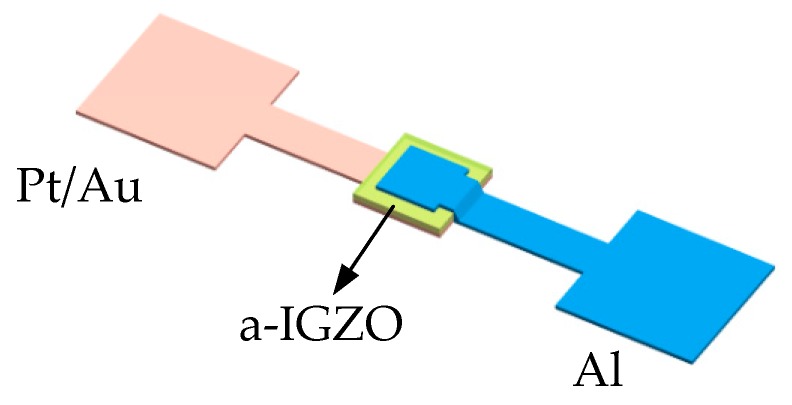
Structure of the a-IGZO Schottky diode. The device consists of a 50-nm-thick Schottky metal layer (pink); an a-IGZO layer (green) with a thickness of 30, 50, 100, or 150 nm; and an Al electrode (blue).

**Figure 2 sensors-19-00224-f002:**
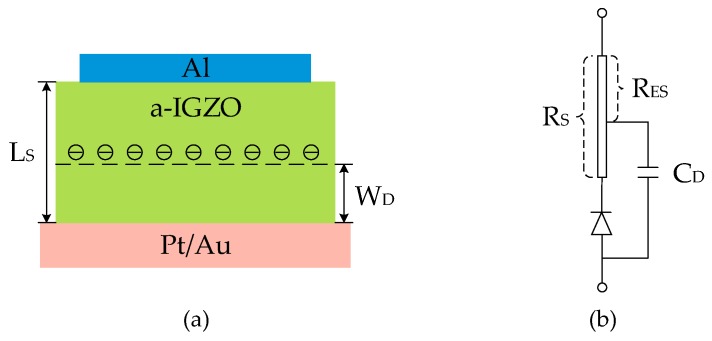
(**a**) Cross-sectional structure of the diode. (**b**) Equivalent circuit of the a-IGZO Schottky diode [27].

**Figure 3 sensors-19-00224-f003:**
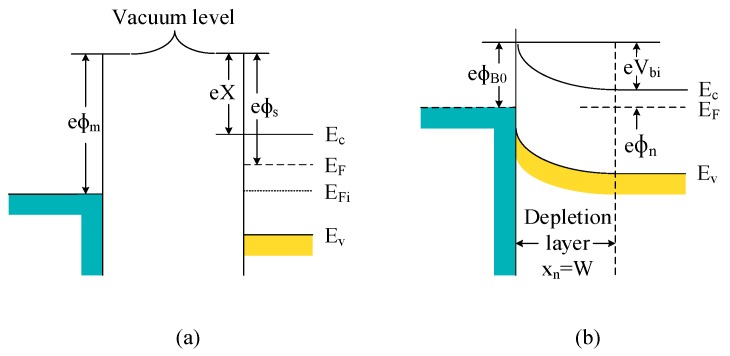
An ideal band diagram of the metal-N semiconductor (**a**) before contact and (**b**) after contact.

**Figure 4 sensors-19-00224-f004:**
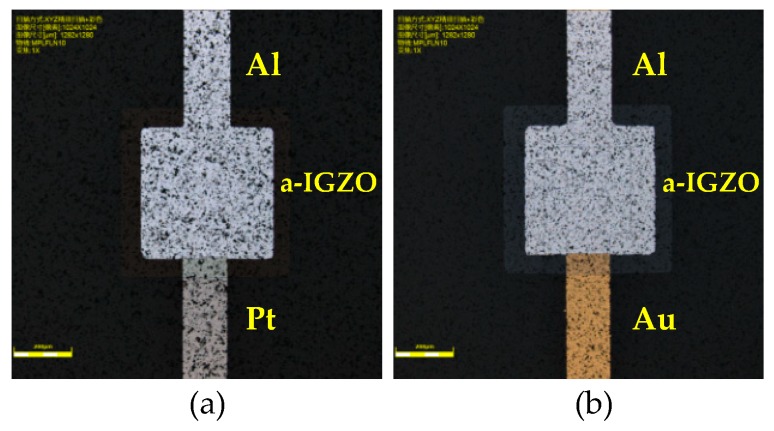
The optical images of the fabricated device. (**a**) The positive electrode material is Pt. (**b**) The positive electrode material is Au.

**Figure 5 sensors-19-00224-f005:**
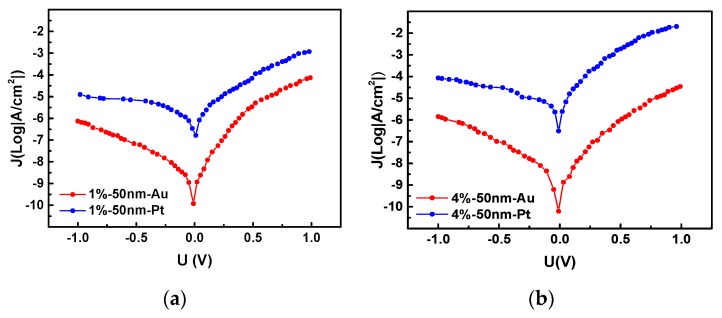
J–V Characteristics of different Schottky metal/a-IGZO diodes. (**a**) Au/a-IGZO and Pt/a-IGZO diodes with sputtering process of O_2_: (O_2_ + Ar) = 1%. (**b**) Au/a-IGZO and Pt/a-IGZO diodes with sputtering process of O_2_: (O_2_ + Ar) = 4%.

**Figure 6 sensors-19-00224-f006:**
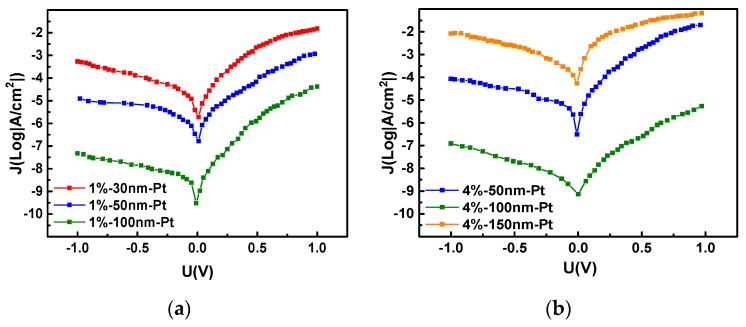
J–V Characteristics of Pt/a-IGZO Diodes with Different a-IGZO Thicknesses. (**a**) J–V characteristics of Pt/a-IGZO diodes with a-IGZO thicknesses of 30, 50, and 100 nm, and sputtering process O_2_: (O_2_ + Ar) = 1%. (**b**) J–V characteristics of Pt/a-IGZO diodes with a-IGZO thicknesses of 50, 100, and 150 nm, sputtering process O_2_: (O_2_ + Ar) = 4%.

**Figure 7 sensors-19-00224-f007:**
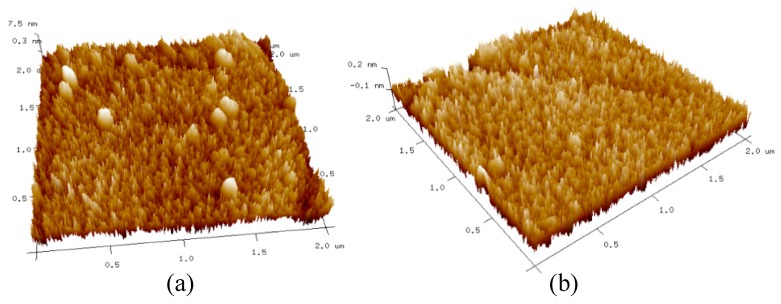
AFM images of 50-nm-thick a-IGZO films in the sputtering process of (**a**) O_2_: (O_2_ + Ar) = 1% and (**b**) O_2_: (O_2_ + Ar) = 4%.

**Figure 8 sensors-19-00224-f008:**
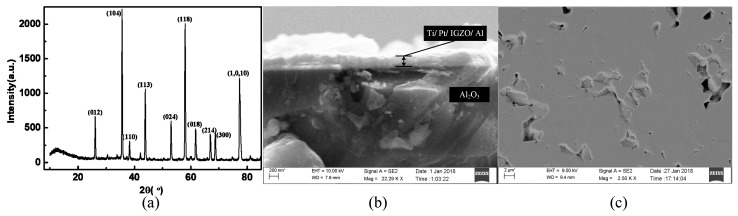
(**a**) XRD pattern of the a-IGZO film on Al_2_O_3_ substrate. SEM image of (**b**) diode cross section and (**c**) a-IGZO film surface.

**Figure 9 sensors-19-00224-f009:**
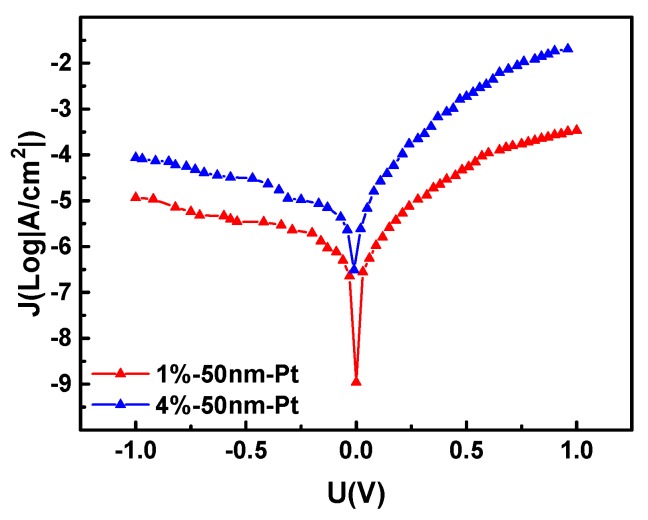
Pt/a-IGZO diodes with a film thickness of 50 nm in a sputtering environment with O_2_: (Ar + O_2_) = 1% and O_2_: (Ar + O_2_) = 4%.

**Figure 10 sensors-19-00224-f010:**
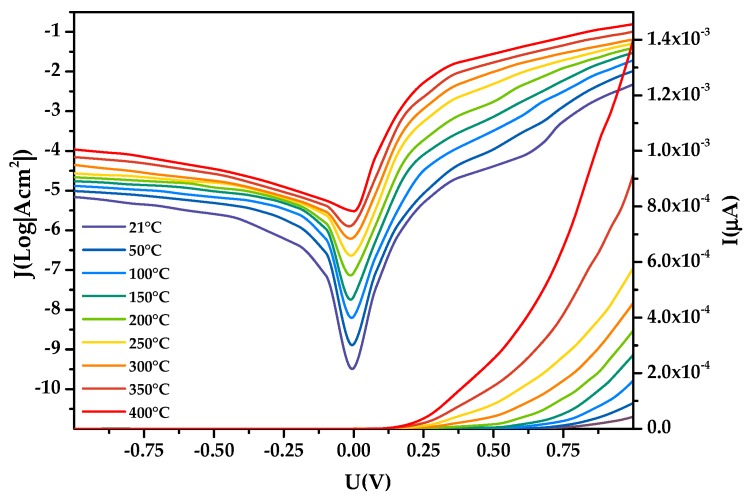
I–V and J-V curve of diode at 21–400 °C.

**Figure 11 sensors-19-00224-f011:**
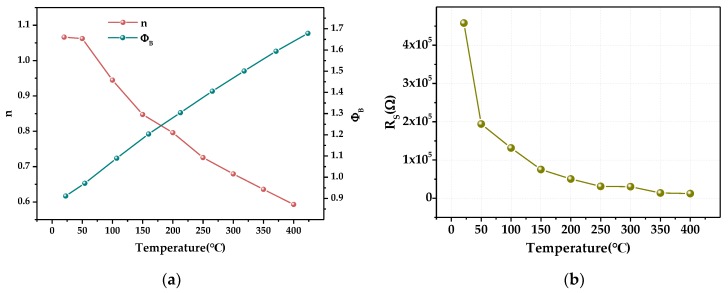
Parameters extracted from Cheung functions. (**a**)The ideality factor *n* and the barrier height Ф*_B_* of the diode at 21–400 °C changes with temperature. (**b**) The resistance of the diode at 21–400 °C changes with temperature.

**Figure 12 sensors-19-00224-f012:**
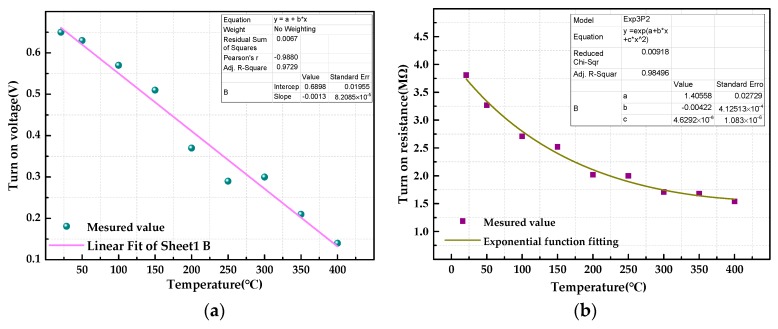
(**a**) Variation in Pt/a-IGZO diode turn-on voltage in the range of 21–400 °C. (**b**) Variation in Pt/a-IGZO diode turn-on resistance in the range of 21–400 °C.

**Figure 13 sensors-19-00224-f013:**
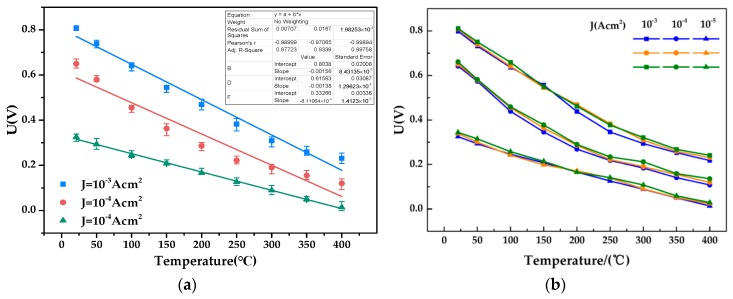
(**a**) Forward voltage versus temperature curve of Pt/a-IGZO diode at 21-400 °C. (**b**) Sensor’s response for temperature at different current density for three cycles.

**Table 1 sensors-19-00224-t001:** EDS Element Composition Analysis of a-IGZO film.

O_2_: (Ar + O_2_)	Element	Element Mass Ratio (%)	Elemental Atomic Ratios (%)
1%	In	28.33	36.12
	Ga	17.34	36.42
	Zn	12.26	27.46
4%	In	12.49	36.03
	Ga	6.93	32.94
	Zn	6.11	31.03

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
