# Peer review of "Al2O3-Based a-IGZO Schottky Diodes for Temperature Sensing"

_sensors, 2019, doi:10.3390/s19020224_

Reviewer 1 Report

The manuscript under consideration describes transport characteristics of different structures designed to operate as Schottky diodes. The structures are based on an amorphous semiconducting material a-IGZO embedded between two metals with different workfunctions. IV-curves of these devices have been measured in a wide temperature range. The obtained experimental data were interpreted within a standard model used for transport through a Schottky diode, i.e. thermionic emission. By itself this applicability of this model is not significant enough to possess any scientific novelty. Authors also claim that presented research allowed for developing a device with properties optimized for use as a temperature sensor. They estimate the best sensitivity as few mV per 0C. If properly supported by experimental data this part will be interesting and useful for the readership of the journal. In the current version of the manuscript the usability of the suggested structures as a t-sensor is unfortunately questionable. There is sufficient amount of information authors have to provide:

- How reproducible is the temperature dependence of the voltage V across a device at fixed bias current. How many heating/cooling cycles do the devices withstand without degrading and how stable are its characteristics. A set of illustrative V(T) curves should be presented in the manuscript

- How reproducible are the properties of the devices for a given technological root

- What is expected accuracy of temperature measurements for different temperature ranges. What are these accuracy estimations based on (noise values, irreproducibility, etc.)

If these data are provided some of the IV curves sets presented in the manuscript may be omitted to preserve the entire volume of the paper. Only optimized structure should be presented in detail. Beside this major modification to the manuscript the authors should take into account the following isuues:

1. Authors should disclose meaning of the term “a-IGZO”, not just IGZO

2. There are several typos in the formulae in the line 118

3. What is IGZO-TFT? ALL abbreviations have to be explained

4. lines 131-135 should be moved to the previous section, devoted to fabrication process

5. Statement “the I–V characteristic curve shifts from right to left when the temperature increases” (lines 228-229) is misleading. Current increase for a given voltage is evident though.

6. How is the “positive turn-on voltage” defined? Is it a voltage corresponding to a certain current or something else is not quite clear. Therefore the temperature dependence of this value is meaningless

Author Response

Dear reviewer,

We all authors thank you for your reviewing work on our manuscript. We have made some modification as you have suggested. The revised manuscript was uploaded, and the revisions were highlighted in Microsoft Word with colored texts.

The specific response to your comments is attached.

Best regards!

    Qianqian Guo

Reviewer 2 Report

 In this paper, the authors fabricate and characterize several IGZO Schottky diodes to select the best device to use as temperature sensor in the range 20-400 °C.

The application and the results are interesting and worth a publication. However, the authors should carefully revise their manuscript, which includes too many typos, grammar errors and contradictions. Also, the analysis of the diode behavior and the extraction of the diode parameters are dubious and need more attention.

The paper can be reconsidered for the publication after a major revision.

Here are comments and concerns that the authors could consider:

Revise the sentence in the abstract: “ Herein, a series of factors influencing diode performance are studied, obtain the relationship between diode ideal  factor n, the barrier height ФB and temperature.” Is it “… are studied to obtain…”?

Lines 47-48 “a-IGZO is a good amorphous semiconductor material, the internal physical properties are isotropic, for a-IGZO with n>1017 cm-3, when the temperature changes, the carrier concentration is basically unchanged, indicating that EF has crossed the mobility side. “ Introduce the EF acronym. “ Clarify the sentence “that EF has crossed the mobility side”.

Lines 50-51, revise the sentence. There is a typo in the definition of the carrier density range: “However, in the range of 1017 cm-3< n< 1016 cm-3, although the carrier concentration is independent of  temperature,..”

“The contact resistance between Al and a-IGZO is 1.191Ωcm2.” – Clarify if this is the measured value in this work.

Lines 102-104: “The flow of electrons in the semiconductor to the lower level of the metal causes the Fermi level to change continuously, and due to the absence of electrons in the semiconductor, positively charged holes form  a space charge region, also known as the space depletion layer.[23]” This is not completely correct. The space charge region is not due to holes but to immobile positive donor ions. Please revise the sentence. See the following reference, which I suggest to cite here in addition to 23: https://doi.org/10.1016/j.physrep.2015.10.003

Lines 110-111 “As represented by equation (2). Vbi is the built-in voltage, and this barrier is formed by the electrons in the conduction band moving toward the metal, similar to the junction barrier.” Revise the punctuation in the sentence. The barrier is not formed “by” the electrons, but rather “for” the electrons.

Line 112-114: “According to the theory of thermionic emission, the forward voltage applied to both sides of the  metal–semiconductor barrier region and the density of the current passing through the rectifying junction can be represented by equation (3)”: I suggest removing the “forward” as formula (3) is valid both in forward and reverse bias.

Lines 131-135: Revise English. “oxygen vacancies are considered to be the main source of IGZO-TFT” This sentence is ambiguous. The paper does not deal with TFTs. Express the concept differently.

I suggest quoting the current density using a more comfortable notation such as yy mA/cm^2 rather than 10^x.xx A/cm^2. Also, pay attention to the significant digits. Obviously, not all the written digits are meaningful if you consider the experimental error (an example “the rectification ratios are 29.242 and 233.884, respectively.” )

“The series resistance has a sensitive dependence on the thickness of the a-IGZO layer.” The authors refer to the series resistance but never explicitly estimate it. They can refer to the following paper for methods to estimates the series resistance as well as the Schottky barrier height or ideality factor of a Schottky diode: ttps://doi.org/10.1088/1361-6463/aac562.

Lines 163-167: “When the a-IGZO film is thin, the film body resistance is small and the carrier concentration increases as the film thickness increases, so the current density increases; when the film thickness is greater than 100 nm, the film bulk resistance is relatively large and the carrier concentration is relatively stable; therefore, the current density shows a decreasing trend. “ Please clarify why the carrier concentration is considered to depend on the IGZO thickness. Can the series resistance dependence be accounted for based only on geometrical factors?

Have the authors considered the voltage drop on the series resistance to fit the measured diode current?

Line 169-172: “ the extracted barrier heights are 0.41, 0.47, and 0.50 eV for diodes with 50, 100, and 150 nm a-IGZO layer thicknesses, respectively. The slope of the linear segment of the curve shows that the diode’s ideal factor n with an a-IGZO film thickness of 50 nm is also the largest.” Give details on how these values are obtained.

I suggest showing the curves of figure 9 on semilog scale over the full bias range, rather than split the curves into two separate plots (inset in figure 9 and figure 11).

Figure 11 is never referenced.

Figure 10: Clarify how the turn-on voltage and the turn-on resistances are defined.

Lines 228-231: “Fig. 9 shows that when the bias voltage is positive, the I–V characteristic curve shifts from right to left when  the temperature increases; the positive turn-on voltage of the Pt/a-IGZO diode increases with the temperature increasing from 21 °C to 400 °C; in addition, the voltage is reduced from 0.64 V to 0.14 230 V, the average voltage drift rate is 1.32 mV/°C, 231”  Is there a typo here? The positive turn-on voltage of the Pt/a-IGZO diode decreases with the temperature.

“Therefore, we choose a region with a large forward bias and good linearity (0.7-1V) for linear fitting. “ The region 0.7-1 V seems to me as the region limited by the series resistance. Therefore, it might be not the best one for the extraction of the barrier height and ideality factor.

“The fitting curves of n and ФB are shown in Fig. 12” Figure 12 does not show the fitting curves, but rather n and phi extracted from the fitting procedure.

The analysis of the diode parameters described at 243-255 is confusing and dubious. It should be fully revised. Please refer to  papers such as https://doi.org/10.1088/2053-1583/4/1/015024 for the use of I-V-T curves for diode parameter extraction.

Line 261-265: give a physical reason for the decrease of resistance and the increase of the barrier for increasing temperature.

Author Response

(The authors gave the same response as above.)

Reviewer 3 Report

This paper presents Al2O3 based indium-gallium-zinc-oxide (IGZO) diode for operation at high temperature environments up to 400 deg C. The reliability of the fabricated diodes was tested with varied IGZO thickness and sputtering oxygen concentration. Interestingly, 50 nm-thick IGZO and 4% of oxygen concentration showed the highest reliability at temperature up to 400 deg C. Considering the importance of wide band gap compounds for harsh environment applications, this work is of interest for researchers in electronics, mechanical, and material science fields. But a few comments below must be addressed before this manuscript is reconsidered in ‘Sensors’

1) ‘1. Introduction’ section: In addition to SiC, gallium nitride (GaN) is the widely used wide-bandgap semiconductor which enables operation at harsh environments [e.g. Diamond Relat. Mater. 18, 884 (2009), J. Phys.: Condens. Matter 16, R961 (2004), Appl. Surf. Sci. 368, 104 (2016), Electronics Lett. 39, 1708 (2003), Appl. Phys. Lett. 108, 012104 (2016)]. Authors need to extend the introduction with extensive references including above papers.

2) ‘2. Design and fabrication’ section: What was the doping type and concentration of IGZO epitaxial layer on top of Al2O3 substrate? More detailed information is needed in this section.

3) ‘3 Results and discussion’ section: Authors need to provide top view images (SEM or optical images) of each fabricated device.

Author Response

Dear reviewer,

We all authors thank you for your reviewing work on our manuscript. We have made some modification as you have suggested. The revised manuscript was uploaded, and the revisions were highlighted in Microsoft Word with colored texts.

The specific response to your comments is attached.

Best regards!

    Qianqian Guo

Round  2

Reviewer 1 Report

Authors have properly responded to all of my criticism except for the most important one. 

- What is expected accuracy of temperature measurements for different temperature ranges. What are these accuracy estimations based on (noise values, issues with reproducibility, etc.)

The problem is that they confuse SENSITIVITY with ACCURACY. Definition of  measurement accuracy can be found, e.g. here https://www.bipm.org/utils/common/documents/jcgm/JCGM_200_2008.pdf

I insist that authors provide experimentally justified value analyzing sources of errors listed above. In case the current devices give unsatisfactory values of accuracy it will not be a reason not to publish the paper. The concept described by the authors may be still useful for certain applications etc.

Author Response

Dear reviewer,

We all authors thank you for your reviewing work on our manuscript again. We have made some modification as you have suggested. The revised manuscript was uploaded, and the revisions were highlighted in Microsoft Word with colored texts.

The specific response to your comments is attached:

Best regards!

Qianqian Guo

Reviewer 2 Report

The authors have made significant changes and improvements in their manuscript and have also corrected some technical errors. They have given convincing responses to the various questions and comments I had raised.

The revised version of the manuscript appears technically sounder. The paper can be accepted for the publication in the current form. 

Author Response

Dear reviewer,

Thank you again for your review and comments and we have a deeper understanding of the theory.

Best regards!

Qianqian Guo

Round  3

Reviewer 1 Report

The manuscript in its present form resolves all the previous issue and I suggest to accept it in the present form